# Treatment of Combined Hepatocellular and Cholangiocarcinoma

**DOI:** 10.3390/cancers12040794

**Published:** 2020-03-26

**Authors:** Simona Leoni, Vito Sansone, Stefania De Lorenzo, Luca Ielasi, Francesco Tovoli, Matteo Renzulli, Rita Golfieri, Daniele Spinelli, Fabio Piscaglia

**Affiliations:** 1Internal Medicine Unit, Department of Digestive Diseases, Bologna Authority Hospital S.Orsola-Malpighi, 40136 Bologna, Italy; 2Department of Medical and Surgical Sciences, University of Bologna, Via Massarenti 9, 40136 Bologna, Italy; vito.sansone@studio.unibo.it (V.S.); luca.ielasi@studio.unibo.it (L.I.); francesco.tovoli2@unibo.it (F.T.); fabio.piscaglia@unibo.it (F.P.); 3Oncology Unit, Department of Experimental, Diagnostic and Specialty Medicine, University of Bologna, 40136 Bologna, Italy; stefania.delorenzo@studio.unibo.it; 4Radiology Unit, Department of Experimental, Diagnostic and Specialty Medicine, Sant’Orsola Hospital, University of Bologna, 40136 Bologna, Italy; matteo.renzulli@aosp.bo.it (M.R.); rita.golfieri@unibo.it (R.G.); daniele.nexus@gmail.com (D.S.)

**Keywords:** hepatocellular carcinoma, intrahepatic cholangiocellular carcinoma, mixed hepatocellular carcinoma-cholangiocellular carcinoma

## Abstract

Combined hepatocellular and cholangiocarcinoma (HCC-CC) is a rare primary liver cancer. It is constituted by neoplastic cells of both hepatocellular and cholangiocellular derivation. Different histology types of HCC-CC have been reported, hinting at heterogeneous carcinogenic pathways leading to the development of this cancer. Due to its rarity and complexity, mixed HCC-CC is a scantly investigated condition with unmet needs and unsatisfactory outcomes. Surgery remains the preferred treatment in resectable patients. The risk of recurrence, however, is high, especially in comparison with other primary liver cancers such as hepatocellular carcinoma. In unresectable or recurring patients, the therapeutic options are challenging due to the dual nature of the neoplastic cells. Consequently, the odds of survival of patients with HCC-CC remains poor. We analysed the literature systematically about the treatment of mixed HCC-CC, reviewing the main therapeutic options and their outcomes and analysing the most interesting developments in this topic with a focus on new potential therapeutic avenues.

## 1. Introduction

Combined hepatocellular and cholangiocarcinoma (HCC-CC) is an uncommon primary liver malignancy, accounting about 2%–5% of primary liver cancers [1,2,3]. The combination of two different histological features characterizes this tumour, which shows elements of both hepatocellular (HCC) and intrahepatic cholangiocellular carcinoma (iCCA). HCC-CC is frequently diagnosed in patients with chronic hepatitis or liver cirrhosis [3,4].

Due to its rarity, this cancer is poorly understood. Consequently, clinical, diagnostic, therapeutic, and prognostic characteristics have not been entirely defined.

In the following paragraphs, we will discuss the clinical presentation, imaging characteristics, histology, and therapy of HCC-CC.

## 2. Diagnosis

Similarly to other types of primary liver cancers, HCC-CC can be diagnosed either symptomatically or during surveillance protocols in patients with chronic liver diseases [5,6,7].

In the first case, the symptoms are not different from those characterizing HCC and iCCA, i.e., fatigue, pruritus, jaundice, and abdominal pain. The final diagnosis is achieved by histologic examination performed on a pre-operative biopsy or the surgical specimen.

In the latter scenario, HCC-CC is found by surveillance imaging performed for an early diagnosis of HCC. As such, patients may be asymptomatic at the diagnosis. Usually, a conventional abdominal ultrasound (US) examination is the first step toward the finale diagnosis. A new nodule found in a cirrhotic liver subsequently prompts a second-level imaging with a contrast medium, including contrast-enhanced ultrasound (CEUS), computed tomography, or magnetic resonance imaging with hepatospecific contrast medium. It should be underlined that all of these tests are primarily performed to distinguish HCC (the most common malignancy in cirrhotic livers) from benign nodules, and not because HCC-CC is suspected. For this aim, specific liver imaging reporting and data system (LI-RADS) classifications have been proposed for the three main diagnostic techniques [8,9]. In the LI-RADS classification, liver lesions are categorized on a scale ranging from LR-1 (definitely benign) to LR-5 (definitely HCC) according to their contrast pattern. In particular, LR-5 lesions show the typical pattern of homogeneous arterial hyper-enhancement followed by a late and mild washout. Lesions showing definite signs of malignancy but patterns that are atypical for HCC (for instance, rim-like arterial enhancement, early or marked washout) are categorized as “LI-RADS metastasis” (LR-M).Usually, LR-M nodules are expressions of undifferentiated HCC, other primary liver cancers (including HCC-CC), or metastasis [8,9]. As these different medical conditions require specific therapeutic strategies, LR-M nodules should undergo a biopsy for a systematic characterization.

HCC-CC shows a range of different contrastographic patterns [10], reflecting its mixed cellular population. In most cases, HCC-CC appears as LR-M nodules, prompting the histology examination and the final diagnosis. However, in a small proportion of cases, HCC-CC can mimic HCC, appearing as an LR-5 lesion. In this unfortunate scenario, the final diagnosis is reached only after surgery, provided that the lesions are resectable. In unresectable cases, this misdiagnosis is particularly difficult to reveal, leading to a possible underdiagnosis of HCC-CC. Usually, in these cases, a not purely hepatocellular nature of the lesion is suspected in the event of a systematic refractoriness to locoregional or systemic treatments, which leads to a tumour biopsy.

## 3. Pathological Classification and Molecular Alterations

According to the World Health Organization (WHO), the histopathological diagnosis of HCC-CC requires the presence of both hepatocellular and biliary components in the tumour [11].

HCC-CC was first reported in 1903 by Wells [12]. Later, the first histological classification was proposed by Allen and Lisa [13] in 1949. In particular, the Allen–Lisa classification included three different histological types of HCC-CC. Different and distinct foci of HCC and cholangiocarcinoma (CC) in the same liver characterized Type A HCC-CC. Colliding masses with features of HCC and CC characterized type B HCC-CC. Type C, instead, was described as a single mass including both hepatocellular and cholangiocellular components.

In 1985, Goodman et al. proposed another classification encompassing three subtypes [3]: type 1 or collision tumour (separate and colliding foci of HCC and CC in the same liver); type 2 or transitional tumour (transitional area of intimate intermingling of the two components with actual transition of HCC elements to CC elements in the same tumour); and type 3 or fibrolamellar tumour (similar to fibrolamellar HCC, but with cells producing mucin). In 1989, the Liver Cancer Study Group of Japan proposed classifying HCC-CC into three subgroups: double cancer, combined type, and mixed type [14].

Recently, the WHO classification reported HCC-CC as a distinct entity and identified two main subtypes: classical type and HCC-CC with stem-cell features [11]. The classical type is characterized by an intermixed area of typical HCC and CC, with the presence of transition foci with intermediate morphology of both types. The stem-cell features type is divided into three subgroups: typical, intermediate, and cholangiocellular. In the presence of predominant stem-cell components, the tumour should be classified as one of the subtypes of HCC-CC with stem-cell features. However, a threshold for the percentage of stem-cell area required for the characterization was not provided.

The histogenesis of HCC-CC remains controversial. Some investigators suggested that this rare cancer might derive from the hepatic progenitor cells (HPCs), a bipotent cell population residing in the terminal biliary ductules with the potential of differentiation into hepatocytes or biliary epithelium cells. HPCs are a heterogeneous cell population expressing phenotypic markers of both immature hepatocytes (such as α-fetoprotein) and bile duct cells (such as bile duct-type cytokeratins) [15,16].

It has been demonstrated that a cell line deriving from a resected HCC-CC could differentiate to both HCC and CC under different growth conditions, suggesting that HCC-CC may arise from stem cells [17,18]. On a similar note, Woo et al. investigated the comparative gene expression profile of CC, HCC, and HCC-CC tumours. They observed that iCCA and HCC could be clearly distinguished from one another according to their gene expression profile [19]. These data were recently confirmed by Xue and colleagues [20] in a large-scale, analysis of 133 Pan-Asian HCC-CC cases. Comparing the mutational frequencies in their cohort of HCC-CC with large datasets of HCC and iCCA, the authors found a higher frequency of TP53 mutations (49%). The frequent HCC-specific mutation of CTNNB1 and the iCCA-specific mutation of KRAS were virtually absent in HCC-ICC [20]. Integration of RNA-seq data and genomic data revealed a fusion gene with potential therapeutic relevance (PTMS-AP1G1) in 11.7% of all cases [20]. Even more interestingly, the authors divided the whole HCC-CC population according to the Allen–Lisa classification, finding that combined (Type B) and mixed HCC-ICCs (Type C) are distinct tumour subtypes in terms of gene expression. Type B tumours were shown to closely resemble iCCA, whereas Type C HCC-CC more closely resembled HCC. Specifically, combined HCC-CCs exhibited increased expression of markers such as KRT19 and PRDM5. In contrast, the HCC-related markers AFP, GPC3, and APOE were highly expressed in mixed HCC-CC [20]. Finally, the authors found that nestin (a protein of the intermediate filaments) was relatively overexpressed in HCC-CC in comparison with other liver cancers, and that nestin+ tumours were characterized by a grim prognosis [20].

## 4. Staging Systems

Different staging systems have been proposed in the last decades to categorize either HCC or iCCA. Most of these staging systems are meant to provide both prognostic information and therapeutic guidance. Examples of such staging systems include the Barcelona Clinic Liver Cancer (BCLC) algorithm [21], the Hong Kong Liver Cancer Staging System for HCC [22], and the Tumour, Node, Metastasis classification(TNM) [23] and other staging systems in the case of intrahepatic CC [24,25].

Until very recently, no staging systems specific for HCC-CC were proposed. Recently, Tian et al. [26] proposed a risk prediction preoperative model to discriminate HCC-CC patients from subjects with other primary liver cancers and to identify the best treatment approach in clinical practice. The authors identified seven different variables as risk predictors for HCC-CC: age, HbcAb, alpha-fetoprotein (AFP), carcinoembryonic antigen (CEA), red blood cells (RBC) count, blood urea nitrogen (BUN), and neoplastic portal vein thrombosis.

## 5. Surgical Treatments

### 5.1. Resection

A surgical approach is the only curative treatment available for patients with HCC-CC. As a consequence, it is the most studied approach for these patients (Table 1).

Both the hepatocellular and cholangiocellular components of this disease can spread within and outside of the liver. Consequently, an accurate perioperative stadiation is mandatory to maximize the benefits derived from a surgical intervention. A major hepatic resection with a hilar lymphadenectomy is the recommended treatment in noncirrhotic patients without extrahepatic spread [31]. In the case of cirrhotic patients, however, hilar lymphadenectomy is performed only in selected patients as it can increase the risk of postoperative ascites. It should be underlined, however, that the role of lymphadenectomy in improving overall survival is still controversial [27,44,45]. A possible confounding factor in this regard is the pre-operative diagnosis of liver cancer. In fact, patients that have a definite histologic diagnosis of not purely hepatocellular malignancy are more likely to be offered a hilar lymphadenectomy. Instead, patients who undergo surgery with the suspicion of HCC and receive the correct diagnosis of HCC-CC only after the pathologic examination of the surgical specimen, are far less likely to have received a lymphadenectomy (which is not routinely performed, as nodal metastasis is a relatively rare event in the setting of small HCC).

A resection margin >10 mm (R0 resection) has been associated with a longer disease-free survival in patients with multifocal disease (40% vs. 0% at 1 year) [43].

Surgery seems to be equally effective in older and in younger patients in terms of survival outcome, suggesting that age alone should not be considered a limiting factor for a major hepatic surgery [46].

In the case of underlying liver cirrhosis, limited hepatic resections may preserve an adequate residual liver function [47]. As seen for other hepatobiliary tumours, an accurate selection of candidates for surgery is also essential.

Despite a curative surgical approach, HCC-CC remains an aggressive tumour with a poor prognosis and is related to a high rate of recurrence and a 5-year survival rate of about 30% [35,41]. With only a few exceptions, [30,34] all of the studies in the literature reported that the postsurgical survival of HCC-CC patients is worse than those with HCC but better than those with CC [28,29,35,41].

This postresectional survival advantage of HCC-CC over CC is also probably related to the availability of locoregional treatments for the early management of recurrence [41].

A prognostic estimation of cholangiocellular-hepatocellular carcinoma after resection (PECAR) score has been recently proposed [48]. The authors identified the following variables as independent predictors of recurrence: male sex, elevated gammaGT, macrovascular invasion, and hilar lymphoid metastasis. The PECAR score consequently ranged from 0 (absence of risk factors) to 4 (all of the risk factors present). The PECAR score successfully predicted 1- and 5-year recurrence risks. However, external validations are needed before definite conclusions can be drawn.

The identification of HCC-CC patients at high risk of recurrence has an important repercussion in terms of adjuvant therapies. It is important to note, however, that effective adjuvant treatments have been elusive both for HCC and iCCA. In the first case, a large Phase 3 trial failed to demonstrate the superiority of sorafenib vs. placebo in the adjuvant setting [49]. In the latter case, international guidelines recently began to recommend capecitabine, despite the results of the BILCAP trial not reaching full statistical significance [50]. Therefore, it can be easily inferred that an effective adjuvant treatment for HCC-CC has not been found to date. Additionally, the relative rarity of this tumour prevents large clinical trials. As a result, the available evidence relies on poor quality data. In the only study that specifically reported the adjuvant treatments performed [43], adjuvant treatments were offered at surgeons’ discretion. Overall, 13 of 50 patients received adjuvant procedures in the form of transarterial chemo- or radio-embolization, systemic chemotherapy, external radiotherapy, molecular targeted therapy, or a combination of any of these. The authors did not find advantages of the adjuvant treatments in terms of recurrence or survival [43].

### 5.2. Transplantation

Liver transplantation has a well-codified role in patients with HCC arising in liver cirrhosis, provided the nodules are within the Milan criteria (single nodule <5 cm; <3 nodules, the largest <3 cm) [6]. Liver transplantation can also be considered in selected patients with iCCA [51].

Data in patients with combined HCC and iCCA (HCC-CC) are scant. No prospective trials have been conducted so far and the whole body of evidence in this regard consists of retrospective data of HCC-CC diagnosed accidentally in explanted livers. To further complicate data analysis, papers about HCC-CC and liver transplantation group together data about HCC-CC and iCCA. Consequently, a thorough analysis of the imaging looking for hallmarks of HCC-CC to correctly diagnose patients on the waiting list is a topic of paramount importance [33].

Overall, the incidence of misdiagnosed or incidental HCC-CC in patients undergoing liver transplantation (LT) was 0.48% (82/17,060) according to an extensive review [52]. In another monocentric series, HCC-CC constituted 2.7% of total LT for liver malignancies [40].

Chan et al. first reported a case series in 2007. Three patients were diagnosed with HCC-CC after LT, two survived with no recurrence 25 and 35 months after transplantation [53]. Maganty et al. subsequently reported another series of patients with worse outcomes: one patient survived 8.5 years after LT, while two died of disease progression 144 and 155 days after LT [54].

Another retrospective study analysed a series of 12 HCC-CC patients who had liver transplants. With a mean follow-up period of 335 days, 58% patients died, 86% of which were for metastases-related causes. Of note, 41% of the patient sample underwent chemotherapy after LT, based on an oncologic consultation. The 1-, 3-, and 5-year cumulative survival probabilities were 79%, 66%, and 16%: outcomes better than iCCA albeit worse than HCC [32].

A cohort study was completed with 3378 HCC patients and 54 HCC-CC patients who underwent resection or transplantation: 35 received surgical resections (64.8%) and 19 received transplants (35.2%). For patients that underwent LT, 1-year and 3-year survival were 89% and 48%, respectively, while those for resected patients were 71% and 46%, respectively. The authors therefore concluded that the survival benefit of LT and resection was similar [36]. A paper by Song et al. showed similar results: 5-year overall survival (OS) for transplanted vs. resected patients was 50% vs. 42.1% [37].

One of the largest population studies to date reported clinical data from a United States database including 465 patients with HCC-CC. Amongst these patients, 55 subjects (13.1%) underwent LT, with a 5-years OS of 41.1%, a disease-specific survival of 52.8%, and no clear survival benefit of LT over surgical therapies. However, it should be noted that 16 out of 55 patients were Milan-out at the time of transplantation [38].

Sapisochin et al. reported data of 14 transplanted patients: 10 patients had HCC-CC (2 had type I according to the Goodman classification, while 10 had type II). No patient received adjuvant therapy. Patients with Goodman type II reported a lower disease-free survival (8 months, range 1.25–67.7 months vs. 23.6 months, range 1.5–85 months) and a higher 5-years risk of recurrence (81% vs. 58%) than patients with Goodman type I and intrahepatic CC. The actuarial OS were shorter in Goodman II patients vs. iCCA or Goodman type I patients (10.3 months, range 1.8–69 vs. 34 months, range 7.1–61.5 months), although the differences were not statistically significant [33].

Another population from the United Network for Organ Sharing (UNOS) database, collected over a large timespan, included 94 patients with mixed aetiology (hepatitis C and alcohol-related cirrhosis were the most represented) who were incidentally diagnosed with HCC-CC after transplantation. The mean OS was 29 months and the 5-year OS was 40%. This figure is similar to that of iCCA (47%), but lower than that of HCC patients (62%). No clear predictors of OS were identified [42].

Factors influencing prognosis were instead described by Wu et al. in a case series of 21 male patients, and included maximum tumour diameter, lymph node metastases, and portal vein invasion [39].

Magistri et al. described three patients transplanted with HCC-CC, all of whom had a recurrence and died. Mean disease-free survival was 7.97 months, while mean OS was 11.7 months [55].

Using a propensity analysis, Jung et al. compared 32 HCC-CC patients (12 of whom had concurrent HCC) who underwent LT with resection. The 1-, 3-, and 5-year survival rates were 85.0%, 80.0%, and 80.0% in patients with HCC-CC alone vs. 83.3%, 63.5%, and 42.3% for patients with a concurrent HCC. Again, no significant OS were observed between transplanted and resected patients. Amongst the prognostic factors, patients with 1-2 HCC-CC ≤2 cm showed acceptable tumour recurrence and overall survival. Concurrent HCC and tumour size >2.5 cm were, instead, predictors of poor survival [40].

Lastly, a meta-analysis and systematic review of the largest set off pooled data available was recently published by Li et al., comparing patients who underwent liver resection and 301 patients who were transplanted. The authors confirmed no significant differences in overall survival and tumour recurrence between the two interventions [56].

Therefore, to date, there is no clear benefit in offering LT to patients with HCC-CC given the non-existent survival benefit and the scarcity of available organs for liver transplantation.

## 6. Locoregional Treatments

Locoregional treatments have been considered in cases of inoperable HCC-CC. Inoperability usually derives from large or multifocal tumours. In this setting, percutaneous treatments are rarely an effective option. Rather, transarterial therapies can be suitable, given their wider range of therapeutic effects. Only a few studies have evaluated the efficacy of these therapies in HCC-CC (Table 2).

The efficacy of transarterial chemoembolization (TACE) has been evaluated in two small retrospective studies [57,59]. The survival outcomes were poorer than those described for HCC. The authors concluded that tumour arterialization was a key factor in predicting response. Since HCC-CC are usually more fibrotic and less vascular than HCC, the benefits of TACE remains debatable. A larger study evaluated the outcomes of transarterial treatments as a whole (TACE, hepatic arterial infusion chemotherapy and transarterial radioembolization) in 18 patients with inoperable HCC-CC from a total cohort of 79 patients [58]. The overall response rate was 47%, with a median OS of 16.0 months. In this case, the authors concluded that transarterial treatments could potentially downstage HCC-CC for surgical treatments and offer a survival benefit.

## 7. Chemotherapy

In a stark contrast with surgery, the role of systemic chemotherapy in HCC-CC remains unclear. Due to the rarity of this tumour, a standard of treatment has not been established, and current evidence relies on case reports and small retrospective studies in which patients were treated according to the guidelines of either HCC or CC [60,61].

Sorafenib is the standard of care for advanced HCC with median overall survival (OS) ranging from 6.5 to 10.7 months [62,63] in the registration trials, the OS is, however, increasing due to the development of second-line treatments and improvements in the management of manifold adverse events [64]. Instead, the combination of gemcitabine and platinum drugs (cisplatin or oxaliplatin) is the standard frontline chemotherapy for CC, with a median OS of 11.7 months with the gemcitabine plus cisplatin (GEM+CDDP) regimen in the ABC-02 trial [65].

In a retrospective multicenter study, Zaanana et al. evaluated the efficacy of GEMOX regimen (gemcitabine + oxaliplatin) in 204 patients with advanced HCC. The median OS was 10 months (95% CI: 9–14), and median progression-free survival (PFS) was 4.5 months (95% CI: 4–6), with an acceptable safety profile [66]. Although HCC has been considered to be chemotherapy-resistant, these data suggested that GEMOX was a potential therapy for advanced HCC because of its lack of renal and hepatic toxicity in cirrhotic patients. In another phase II study, Zhu et al. investigated the role of bevacizumab (a vascular endothelial growth factor antibody) in association with gemcitabine and oxaliplatin in the treatment of both HCC and CC patients [67,68]. However, both of these studies included only a few patients, and these findings must be confirmed by further studies.

In the literature, there are few reports of systemic chemotherapy for HCC-CC, including single cases or case series [60,69,70,71,72,73]. Recently, three retrospective studies evaluated the efficacy of different treatment regimens in the first-line setting, as shown in Table 3.

In a French multicenter retrospective study, 30 patients with advanced HCC-CC received GEMOX (n = 18, 60%), GEMOX plus bevacizumab (n = 9, 30%) or GEM+CDDP (n = 3, 10%). In patients treated with gemcitabine combined with cisplatin or oxaliplatin, the median PFS and OS were 9.0 months and 16.2 months, respectively, in line with the results obtained in CC patients [74]. There were no differences in outcomes between the three treatment regimens. However, no conclusions can be drawn from this study because of its retrospective nature and its small sample size.

Another retrospective study evaluated 36 patients with advanced HCC-CC treated with systemic chemotherapy. The first line treatment included GEM+CDDP (n = 12), gemcitabine plus 5-fluorouracil (GEM+5-FU, n = 11), sorafenib (n = 5), and other regimes (n = 8, including S-1 monotherapy, gemcitabine monotherapy, and gemcitabine plus S-1) [61]. The median OS of GEM+CDDP, GEM+5-FU, sorafenib, and other groups was 11.9, 10.2, 3.5, and 8.1 months, respectively. There were no statistical differences in median PFS between the four groups. With the multivariate analysis, the authors showed that the OS of patients treated with sorafenib was inferior compared to that of patients treated with platinum-containing regimens (HR: 15.83 [95% CI: 2.25–111.43], *p* = 0.006).

To date, the report of Trikalinos et al. is the most extensive study in the literature. This study included a total of 123 HCC-CC patients [75]. Among these patients, 68 had metastatic disease and were treated with systemic therapy: 16 patients received gemcitabine alone or in combination with 5-fluoropyrimidine (GEM+5-FU); 41 patients received gemcitabine plus a platinum drug (cisplatin or oxaliplatin); 7 patients received sorafenib monotherapy, and four patients were treated with other drugs. In the first two groups, the median OS was 11.7 months and 11.5 months, respectively, while the median OS in the sorafenib group was lower compared to that of the gemcitabine-group (9.6 months). Median PFS was 8.0 months in the gemcitabine/platinum group and 6.6 months in GEM+5-FU group, without reaching a statistical significance (*p* = 0.33). For sorafenib monotherapy, PFS was only 4.8 months.

Taken together, all the available evidence suggests that platinum-containing regimens are the most promising first-line therapy for patients with unresectable HCC-CC, while sorafenib monotherapy appears to be far less effective against this tumour. However, prospective studies are needed to confirm these findings.

## 8. Conclusions

Combined HCC-CC is characterized by both rarity and extreme heterogeneity. This unfortunate combination has prevented the identification of standardized treatments. In particular, the lack of codified systemic treatments is a striking characteristic of HCC-CC, differentiating it both from HCC and ICC. As a consequence, there are a lot of unmet needs both for patients with recurrence after surgery and for patients in the advanced stages. The possibility of addressing these needs in the future relies on the identification of more combined HCC-CC and on understanding its heterogeneous characteristics.

The first task appears particularly daunting, as it deals with a natural rarity of this neoplasm. However, it should be underlined that the correct identification of HCC-CC has been severely hindered by the avoidance of biopsies in patients with liver cirrhosis and neoplastic nodules with an imaging pattern typical for HCC [10]. In fact, the possibility of combined HCC-CC appearing as a typical HCC has been reported. This scenario will likely change in the next years as the need for liver biopsies will likely increase due to the increasing number of clinical trials and therapeutic options requiring the identification of biomarkers [76]. Therefore, an increase in the number of identified combined HCC-CC is also likely.

The second task is of particular interest. The morphological heterogeneity of combined HCC-CC, defined since the time of the Allen–Lisa classification, mirrors a deep inter- and intratumor genetic heterogeneity. The current availability of next-generation sequencing (NGS) techniques able to sequence large quantities of genomes in minimal time will probably help in this task. The intriguing study of Xue et al. [20] is a typical example of the potential of these novel techniques. By demonstrating that Type B HCC-CC has a gene expression similar to iCCA while the Type C HCC-CC genetic signature resembles HCC, the hypothesis of different treatments according to histotype gained support. However, more data are needed to confirm this hypothesis and to verify whether lineage transdifferentiations occurring under cancer therapies might represent a therapy-resistance mechanism [77], as seen in other tumours [78].

Some novel therapies might come even earlier thanks to the therapeutic advances in both HCC and ICC. Immune checkpoint inhibitors (ICI) are under investigation for HCC [79] and ICC [80]. In HCC, a combination of the ICI atezolizumab and the anti-VEGF bevacizumab has recently shown superiority to sorafenib as a frontline systemic treatment in a global Phase 3 trial, also showing a manageable safety profile [81]. Data about ICIs in ICCa are scarce. However, encouraging results of pembrolizumab have been reported in the cohort of the Phase 1 KEYNOTE-028 trial, which enrolled patients with biliary tract cancers. These data prompted the enrolment of a successor cohort of 100 biliary cancer patients in the ongoing KEYNOTE-158 trial (NCT02628067), as well as a number of other trials investigating different ICIs [80]. Should the immune-oncology drugs demonstrate efficacy both in ICCa and HCC, there would be a strong rationale for using these drugs in combined HCC-CC as well.

ICIs are not, however, the only drugs that have recently shown both efficacy and a good safety profile in this field. Ivosidenib, an oral inhibitor of isocitrate dehydrogenase 1 (IDH-1), reached its primary endpoint of increasing the progression-free survival in comparison with placebo in patients with previously treated ICCa [82]. These results testify to the importance of the identification of drug targetable neoplastic polymorphisms and pave the way for the use of more targeted and less toxic medications even for ICCa. If this trend continues, as it appears likely, it will probably become possible to simultaneously treat both the cholangiocellular and hepatocellular components of combined HCC-CC. Currently, this possibility is theoretically possible but de facto prevented by the risk of high-grade toxicities of a combination of gemcitabine-based regimens and sorafenib.

In conclusion, the relative rarity of combined HCC-CC has hindered the research of effective treatments for this condition. However, technological progress and the parallel novel therapies in the fields of HCC and ICCa will probably improve the treatment of HCC-CC in the next few years.

## Figures and Tables

**Table 1 cancers-12-00794-t001:** Main studies of surgery in patients with combined hepatocellular and cholangiocarcinoma.

Authors	Reference	Study Design	Patients	Surgical Treatment		Median OS (Months)	5-year OS (%)	Median DFS (Months)	5-year DFS (%)	Total Recurrence (%)
Nakamura et al., 1996	[27]	Case series	6	Limited resection	4 (66.7%)	52.5	60.0	n/a	n/a	4 (66.7)
				Major resection	2 (33.3%)					
Koh et al., 2005	[28]	Retrospective	24	Hepatic resection		37	n/a	n/a	n/a	14 (58.3)
Lee et al., 2006	[29]	Retrospective	33	Limited resection	10 (30.3%)	47.3	n/a	23.4	n/a	16 (48.5)
				Major resection	22 (66.7%)					
				Liver transplantation	1 (3%)					
Zuo et al., 2007	[30]	Retrospective	15	Limited resection	7 (46.7%)	n/a	8.0	n/a	n/a	n/a
				Major resection	8 (53.3%)					
Kim et al., 2009	[31]	Retrospective	29	Limited resection	12 (41.4 %)	28.8	n/a	n/a	n/a	18 (62.1)
				Major resection	13 (44.8%)					
				Liver transplantation	4 (13.8%)					
Panjala et al., 2010	[32]	Case series	12	Liver transplantation		43.2	16.0	17.7	n/a	7 (58)
Sapisochin et al., 2011	[33]	Retrospective	14	Liver transplantation		19.5	n/a	8	n/a	8 (57)
Lee et al., 2011	[34]	Retrospective	30	Limited resection	24 (80%)	18.3	n/a	5.4	n/a	26 (86.6)
				Major resection	6 (20%)					
Yin et al., 2012	[35]	Retrospective	113	Hepatic resection	103 (91.2%)	16.5	36.4	8	n/a	67 (65)
				Other	10 (8.8%)	/	/	/		/
Groeschl et al., 2013	[36]	Retrospective	54	Hepatic resection	35 (65%)	36	28.0	n/a	n/a	n/a
				Liver transplantation	19 (35%)					
Song et al., 2013	[37]	Retrospective	76	Hepatic resection	68 (89.5%)	n/a	42.1	n/a	26.2	50 (73.5)
				Liver transplantation	8 (10.5%)		50.0		37.5	
Garancini et al., 2014	[38]	Retrospective	465	Limited resection	35 (7.5%)	n/a	10.5	n/a	17.8	n/a
				Major resection	47 (10.1%)					
				Liver transplantation	61 (13.1%)					
				Other	322 (69.2%)					
Wu et al., 2015	[39]	Retrospective	21	Liver transplantation		23	39.0	n/a	30.0	8 (38.1)
Jung et al., 2017	[40]	Retrospective	132	Hepatic resection	100 (75.7%)	n/a	63.0	n/a	n/a	42 (42)
				Liver transplantation	32 (24.3%)		66.0			12 (38)
Yoon et al., 2016	[41]	Retrospective	53	Limited resection	31 (58.5%)	~24	30.5	~7	n/a	39 (75)
				Major resection	22 (41.5%)					
Vilchez et al., 2016	[42]	Retrospective	94	Liver transplantation		29	40.0	n/a	n/a	n/a
Ma et al., 2017	[43]	Retrospective	42	Limited resection	13 (30.9%)	32	35.4	9	23.6	33 (78.6)
				Major resection	29 (69.1%)					

**Table 2 cancers-12-00794-t002:** Reported cohorts of patients treated with locoregional therapies.

Authors	Reference	Study Design	Patients	Treatment		Median OS (Months)	3-year OS (%)	Median PFS (Months)	5-year PFS (%)	Tumour Progression (%)
Kim et al., 2010	[57]	Retrospective	50	TACE		12.3	16.0	n/a	n/a	15 (30)
Fowler et al., 2015	[58]	Retrospective	79	Locoregional	TACE (6)	8.3	n/a	16	n/a	2 (30)
					TARE (6)					3 (50)
					HAI pump (6)					0
				Surgery	33	Not reported		Not reported		15 (44)
				Chemotherapy	28	Not reported		Not reported		Not reported
Na et al., 2018	[59]	Retrospective	42	TACE		32.6	n/a	3.4	n/a	37 (88.1)

OS: overall survival; PFS: progression-free survival; TACE: transarterial chemoembolisation; TARE: transarterial radioembolisation; HAI: hepatic artery infusion

**Table 3 cancers-12-00794-t003:** Literature review of systemic chemotherapy regimens in the first-line setting.

Authors	Reference	Study Design	N° Patient(s)	Regimen (1st Line)		Median OS (Months)	Median PFS (Months)
Hatano et al., 2009	[69]	Case report	1	S-1		n/a	n/a
Kitamura et al., 2008	[70]	Case report	1	5-FU + CDDP		6	/
Shimizu et al., 2009	[71]	Case report	1	UFT		14	/
Kim et al., 2010	[72]	Case report	1	Sorafenib		/	2
Chi et al., 2012	[73]	Case report	1	GEM + CDDP		/	12
Rogers et al., 2017	[60]	Case series	7	GEM	1 (14.3%)	/	3.6
				GEM + Beva	2 (28.6%)	/	4.8
				GEM + CDDP	1 (14.3%)	/	17.0
				Sorafenib	3 (42.8%)	/	2.7
Salimon et al., 2018	[74]	Retrospective	30	GEM + OX	18 (60%)	16.2	9.0
				GEM + OX + Beva	9 (30%)		
				GEM + CDDP	3 (10%)		
				GEM + OX	18 (60%)		
Kobayashi et al., 2018	[61]	Retrospective	36	GEM + CDDP	12 (33.3%)	10.2	3.0
				5-FU + CDDP	11 (30.5%)	11.9	3.8
				Sorafenib	5 (13.8%)	3.5	1.6
				Other	8 (22.2%)	8.1	2.8
Trikalinos et al., 2018	[75]	Retrospective	68	GEM + platinum drug (CDDP or OX)	41 (60.3%)	11.5	8.0
				GEM ± 5-FU	16 (23.5%)	11.7	6.6
				Sorafenib	7 (10.3%)	9.6	4.8
				Other	4 (5.9%)	n/a	n/a

S-1: tegafur/gimeracil/oteracil; 5-FU: 5-fluorouracil/leucovorin; CDDP: cisplatin; UFT: tegafur/uracil; GEM: gemcitabine; OX: oxaliplatin; Beva: bevacizumab.

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
