# Peer review of "Treatment of Combined Hepatocellular and Cholangiocarcinoma"

_cancers, 2020, doi:10.3390/cancers12040794_

Round 1
Reviewer 1 Report
This is a nice narrative review on the treatment of mixed HCC-ICC. The topic is interesting to surgical oncologists and HPB surgeons since patients diagnosed with a mixed HCC-ICC type usually have poor prognosis and worse outcomes compared with pure HCC patients. I congratulate the authors on putting together the existing literature to provide an assessment and an overview of the treatment approaches and associated outcomes among patients with mixed HCC-iCC. I have the following suggestions.
1) Extensive editing of English is required throughout the entire manuscript. There are many sentences with grammatical errors that make the manuscript hard to follow.
2) The authors should include a table with the largest cohorts reporting on mixed HCC-ICC and associated outcomes following surgery. Since surgery is the most effective curative treatment option for patients with resectable disease, it is unknown why the authors chose to present only a table with systemic chemotherapy. I strongly recommend adding one such table
3) Can the authors comment on the utilization of adjuvant chemotherapy following surgery for mixed HCC-ICC?
4) Can the authors comment on the use of lymphadenectomy among patients with mixed HCC-ICC when surgery is performed?
Reviewer 2 Report
General comments
Leoni et al. summarized the latest knoweledges about mixed haptocellular-cholangiocellular carcinoma in this review articel. In spite of very limited information, the authors attempted to overview the literature.
- Page 1, line 38. “more frequently” in patients with cirrhosis compared to what?
- et al. is needed to convert to ilatic.
- Table with the summary of all possible treatments as well as chemotherapy would make it easy to overview the current status of treatment options.
- The part of diagnosis looks scarce. What is LI-RADS? What is the histological diagnosis? What is the typical histology of HCC-CC? Ideally H&E staining images should be included.
- Description about the molecular alterations would also be informative (e.g. Xue et al. Cancer Cell 2019).
Reviewer 3 Report
Thank you for giving me the opportunity to review this very interesting paper about a rare disease entity. Its a comprehensive review of the available literature.
A few comments, a more widely used term for biliary tract cancers is cholangiocarcinoma rather cholangiocellular carcinoma. Also the paper should be edited for grammar and syntax prior to publication, including consistent UK or US spelling. Finally, in Table 1, explain what are the numbers and percentages next to the regimen, in the heading it states 1st line but the total # of patients doesn't match.
Reviewer 4 Report
This is a review article regarding treatment of mixed hepatocellular-cholangiocellular carcinoma. The manuscript is well written, but the authors should answer several raised concerns.
- The authors should comment on pretreatment diagnosis, as treatment strategy may depend on pretreatment diagnosis. For example, if the pretreatment diagnosis is HCC, hilar lymphadenectomy is not performed. Similarly, the choice of chemotherapy regimen depends on whether the pretreatment diagnosis is HCC or ICC.
- Is treatment strategy different between pathological types of HCC-CC?
Round 2
Reviewer 1 Report
All points were addressed
Reviewer 4 Report
The authors revised their manuscript according to reviewer's comments.